# Physical Drivers of the November 2023 Heatwave in Rio de Janeiro

Catherine C. Ivanovich[1], Adam H. Sobel[1,2,3], Radley M. Horton[2,4], Ana M. B. Nunes[5], Rosmeri Porfírio da Rocha[6], and Suzana J. Camargo[2,4]

[1]Department of Earth and Environmental Sciences, Columbia University, New York, NY, United States
[2]Lamont-Doherty Earth Observatory, Columbia University, Palisades, NY, United States
[3]Department of Applied Physics and Applied Mathematics, Columbia University, New York, NY, United States
[4]Columbia Climate School, Columbia University, New York, NY, United States
[5]Instituto de Geociências, Universidade Federal do Rio de Janeiro, Rio de Janeiro, RJ, Brazil
[6]Instituto de Astronomia, Geofísica e Ciências Atmosféricas, Universidade de São Paulo, São Paulo, SP, Brazil

*Correspondence to*: Catherine Ivanovich (cci2107@columbia.edu)

**Abstract**
As extreme heat has not historically been a major hazard for the city of Rio de Janeiro, the
November 2023 Heatwave magnitude and timing were staggering. Here we conduct a case study
of reanalysis data and high-resolution projections to explore the event drivers and characterize
the evolving extreme heat risk in the city of Rio de Janeiro. We find that the heatwave was
associated with atmospheric blocking, potentially linked to the 2023-24 El Niño event. Soil
moisture declines increased surface sensible heat flux, and elevated sea surface temperatures
reduced coastal cooling. The heatwave was preceded by weeks of suppressed precipitation and
terminated by the onset of rain. We also find a significant historical increase in the frequency of
high heat days throughout Brazil and a lengthening of the heat season in the city of Rio de
Janeiro. The frequency of the city's austral spring heat extremes is expected to increase further in
the future, highly dependent upon our future emissions pathway. These results emphasize the
rapidly emerging risk for extreme heat in the city of Rio de Janeiro.
**1 Introduction**

In the spring of 2023, the city of Rio de Janeiro experienced a high impact heatwave that

caught the world's attention. Media sources ranging from local reporting to international news
companies centered stories on the event's record-breaking temperature magnitudes and
unseasonal timing, arriving earlier in the warm season than typical heatwaves (Correio
Braziliense, 2023; Hughs and Jeanet, 2023). The impacts of the extreme heat were widely
publicized in part due to the tragic death of a concertgoer hospitalized during a Taylor Swift
performance in Rio de Janeiro on November 17, with news articles reporting heat-induced
cardiovascular distress as the cause of death (Nguyen, 2023). Sources also report that the stadium
in which the concert took place experienced higher temperatures than those measured in the open
air, as well as a lack of cooling equipment and insufficient water for attendees (Nguyen, 2023;
Jornal Nacional, 2023). Such complexities highlight that extreme heat experienced by
individuals on the ground can far exceed temperatures measured at local weather stations,
depending on infrastructure and the capacity for cooling interventions (Wilby et al., 2021; Nahlik
et al., 2017). However, the meteorological event itself is of course one of the preconditions for
societal impacts. We therefore explore the physical mechanisms behind the heatwave as one key
step towards improving preparation for the impact of future extreme heat events.

Throughout Brazil, the highest temperatures occur climatologically in low latitude and

low altitude regions in the interior of the country, such as the cities of Teresina (Piauí, in the
Northeast region of Brazil) and Palmas (Tocantins, in the Central-West region of Brazil; Alvares
et al., 2013), both of which are far from Rio de Janeiro. Extreme temperatures tend to be
intensified by land-atmosphere interactions, as dry soils partition more energy into sensible heat
(Geirinhas et al., 2018). These relationships between the atmosphere and land surface processes
increase the likelihood of compound extreme heat and drought events and intensify impacts on
agriculture (Cirino et al., 2015), worker productivity for outdoor laborers (Bitencourt et al.,
2021), wildfire risk (Libonati et al., 2022), and direct impacts on human health (Zhao et al.,
2019). As the frequency and intensity of extreme heat throughout Brazil has increased
significantly in the past decades and is projected to continue in the future (Feron et al., 2019;
Regoto et al., 2021; Bitencourt et al., 2020), the widespread socio-economic impacts of these
events are likely to grow.

While Rio de Janeiro is the second most populous city in Brazil (Instituto Brasileiro de

Geografia e Estatística 2022) and the third most populous city in South America (United Nations
Department of Economic and Social Affairs Population Division 2022), few studies have
focused on extreme heat in the city. On one hand, Rio de Janeiro has not historically been a
major hotspot of extreme heat in Brazil and has experienced fewer heatwaves relative to other
major cities in the country (Geirinhas et al., 2018). Further, the numerous microclimates within
the city, influenced by its coastal setting and complex topography, complicate the study of local
heatwave dynamics. Indeed, there is large spatial variability in temperature extremes across the
Rio de Janeiro metropolitan area compared to other Brazilian cities (Alvares et al., 2013).
However, impactful heatwaves in recent decades have increasingly drawn attention from public
health officials and scientific communities alike. Recent literature has explored the dynamics and
mortality impacts of extreme temperatures during heatwaves in 2010 (Geirinhas et al., 2019) and
2013/2014 (Geirinhas et al., 2022) and has begun to investigate compound heatwave and drought
events throughout Southeast Brazil (Geirinhas et al., 2021). There is also building evidence that
temperature extremes are increasing in intensity and frequency throughout Brazil, including the
city of Rio de Janeiro (Regoto et al., 2021; Bitencourt et al., 2019). Climate variability also plays
an important role in modulating temperatures over this area, including large scale modes of
climate variability such as the El Niño-Southern Oscillation (Rehbein and Ambrizzi 2023; Cai et
al., 2020; Shimizu and Ambrizzi 2015), the Pacific Decadal Oscillation, and the Atlantic
Multidecadal Oscillation (He et al., 2021). Should more intense, frequent, and unseasonably
early extreme heat events take place in the future in the city of Rio de Janeiro, these heatwaves
may have increased impacts on human health due to potential exceedance of unprecedented
temperature thresholds and individuals' lack of preparation for these events. In a tropical city
where baseline temperatures are already relatively high, small shifts in the temperature
distribution can have large impacts on the frequency of extremes (Cheng et al., 2019),
particularly at thresholds relevant to human health outcomes (Vecellio et al., 2022). These health
risks are compounded by the humidity in Rio de Janeiro, a coastal city with ample moisture
sources from the ocean and surrounding vegetation, priming the region for humid heat extremes
which are physiologically more dangerous to human health than dry heat (Mora et al., 2017).

In this study, we explore the meteorological conditions that led to the extreme heat event

in November 2023 in the city of Rio de Janeiro. We identify drivers of the exceptional
magnitude and persistence of the extreme temperatures, as well as their early arrival in the
calendar year. We compare these conditions to those associated with typical heatwaves in the
region, and particularly events taking place in the spring season. We then consider how extreme
spring temperature events have shifted throughout the historical period, and how we might
expect them to change in the future with ongoing anthropogenic climate change.

**2 Methods**
**2.1 Data**

This analysis employs both station-based observations and reanalysis data. Initial

analyses are conducted on subdaily station data from the city of Rio de Janeiro, accessed via the
Met Office Hadley Center's HadISD station-based dataset (Dunn 2019) and the Rio Alert
System produced by the Rio de Janeiro City Hall (Sistema Alerta Rio da Prefeitura do Rio de
Janeiro 2024). Three airport weather stations are available from HadISD for the city of Rio de
Janeiro, namely the Galeão/Antonio Carlos Jobim International Airport (located on the island
Ilha do Governador within the Guanabara Bay), the Campo Délio Jardim De Mattos Airport (an
Air Force base located in the city's North Zone), and the Santos Dumont Airport (a waterfront
airport located near the city center). Six additional stations from the Rio Alert System dataset
record measurements from the tops of various community and commercial buildings, including
hotels, schools, and warehouses. These stations are located in distinct areas of the city, whose
topographical and coastal complexities contribute to various microclimates. These stations thus
record distinct values, both instantaneously and on average (see Fig. S1), a challenge that has
been previously identified in the literature (Lyra et al., 2018; Dereczynski et al., 2013). We
therefore base the majority of our analysis on reanalysis data and compare the identified patterns
with station data when possible. This comparison is particularly important for extreme events, as
the magnitude of extreme heat has been shown to be biased in reanalysis products due to their
spatial and temporal smoothing of observations (Rogers et al., 2021; Raymond et al., 2020).
Further, the human experience of heat stress is inherently hyperlocal, meaning that the distinct
microclimates existing throughout the city can control heat stress exposure and the efficiency of
adaptation strategies. However, the present study is primarily concerned with the regional drivers
of the extreme event rather than its absolute magnitude. Reanalysis provides continuous spatial
coverage and a wide array of internally consistent meteorological variables, which warrants its
use for the application here.

Hourly meteorological data are retrieved from the fifth major global reanalysis of the

European Centre for Medium-Range Weather Forecasts (ERA5), including 2-meter temperature,
2-meter dewpoint temperature, volumetric soil water for layer 1 (0-7 cm, where the surface is at
0 cm), surface pressure, geopotential height at 500 hPa and 200 hPa, precipitation, evaporation,
2-meter horizontal winds, and vertical velocity at 500 hPa (Hersbach et al., 2020). From this
hourly data, daily maximum temperature, daily total precipitation, and daily means of all other
variables are calculated from 1979-2023. Daily mean sea surface temperature (SST) data from
1979-2023 is also retrieved from the NOAA 1/4° Daily Optimum Interpolation Sea Surface
Temperature (OISST) dataset (Huang et al., 2021).

We also explore the future evolution of temperature extremes over the city of Rio de

Janeiro using the NEXGDDP dataset (Thrasher et al., 2022). This data product is statistically
downscaled from the Coupled Model Intercomparison Project Phase 6 (CMIP6) models, with a
spatial resolution of 0.25 degrees and outputs variables on a daily temporal scale. We directly
retrieve daily maximum temperature data through the end of the century under the Shared
Socioeconomic Pathways (SSPs) SSP2-4.5 and SSP5-8.5 for the 23 models which output this
variable and pair of scenarios for each day in the calendar year through 2100.

Because of Rio de Janeiro's complex coastal and mountainous terrain, projections may

not accurately capture fine scale differences in the city's climate. For example, recent literature
has shown that the coastal cooling relative to inland areas experienced in regions such as the
eastern United States may be underestimated by global climate models (Raymond and Mankin
2019). These biases are greatest in regions with large land-ocean surface temperature contrasts,
however, and Rio de Janeiro's location in the tropics, as well as the fact that the extreme events
analyzed in this study take place in the spring when this temperature gradient should be
relatively small, may mute these biases compared to other regions and seasons. In order to
address these potential sources of error, we generate a set of synthetic time series based on
NEXGDDP projections which retain the seasonality and variability recorded in the historical
reanalysis data from ERA5. We use a percentile matching technique in which we first bin all
data for the grid cell which includes the city of Rio de Janeiro during a historical base period
(1981-2013) into one-percentile bins for both the NEXGDDP and ERA5 datasets. We
additionally bin all NEXGDDP data from this grid cell into one-percentile bins during one
midcentury period (2041-2060) and one end-of-century period (2081-2100). We then calculate
the temperature delta for each percentile bin between the base period and both the midcentury
and the end-of-century periods in the NEXGDDP data. Finally, we add these percentile-specific
change factors to every data point in each associated bin in the historical ERA5 base period.

**2.2 Methodology**

We first create time series for the historical day-of-year climatologies of variables in the

city of Rio de Janeiro and compare them to the evolution throughout 2023. All anomalies are
calculated relative to historical mean calendar date values (i.e., the daily maximum temperature
anomaly on November 18, 2023 is calculated by subtracting the mean daily maximum
temperatures on November 18 in all previous years in the historical record from the recorded
absolute magnitude of the event). We also generate maps of concurrent meteorological variables
relevant to the extreme heat event for the greater region outside of Rio de Janeiro. We compare
these spatial patterns to those experienced during previous extreme heat events in Rio de Janeiro,
calculated as 99[th] percentile daily maximum temperature days across all seasons for the grid cell
which includes the Galeão International Airport weather station. We then select for events that
occur in the September-November (SON) austral spring season.

We also quantify how extreme heat in the city of Rio de Janeiro is shifting using a variety

of methods. We first calculate the trend in the frequency of extreme temperatures over the
historical record in Brazil, defining these extreme temperatures using both absolute and relative
thresholds. We select these thresholds as 30°C and the locally defined 90$^{th}$ percentile daily
maximum temperature at each grid cell. These thresholds are chosen in order to investigate
impactful temperature magnitudes while ensuring sufficient sample size for the trend analysis.

We also visualize the broadening of the extreme heat season, calculated based on the

number of days between the start of the first heatwave and end of the last heatwave of the
season. A heatwave is defined here as a three-day period with consecutive daily maximum
temperatures above the 50$^{th}$ percentile of daily maximum temperatures across the two hottest
months of the year in the city of Rio de Janeiro (January and February); this 50$^{th}$ percentile
threshold is equal to about 31.4°C. This heat season definition is informed by a definition used
by the United States Environmental Protection Agency (US EPA 2021), adapted to better reflect
Rio de Janeiro's lower temporal variability in temperature due to its tropical location.

Finally, we calculate how spring temperature distributions have already changed in Rio

de Janeiro by comparing early and late historical periods in ERA5 for the grid cell which
includes the Galeão International Airport weather station. Distributions are calculated from
annual spring maximum temperatures in the city of Rio de Janeiro and fit using GEV
distributions, which have been shown to well capture extreme temperature distributions (Powis
et al., 2023; Van Oldenborgh et al., 2022). For comparison, we also plot GEV distributions for
early and late historical periods in the NEXGDDP model data before applying our bias-
correction technique. The location parameter and spread of the model data distributions during
these periods is much lower than that of ERA5 (Fig. S2), further motivating our use of synthetic
time series to explore how these distributions may change in the future.

We then use the bias-corrected NEXGDDP data for the 23 models which report daily

maximum temperature for each day in the calendar year under the aforementioned SSP2-4.5 and
SSP5-8.5 scenarios during a midcentury and end-of-century time period. We additionally
evaluate the impact of only using models which most accurately reproduce the historical
observed daily maximum temperature record in the city of Rio de Janeiro. We calculate the
Perkins skill score to evaluate the similarity between probability density functions of daily
maximum temperature in the reanalysis dataset (ERA5) and each of the 23 global climate models
during the historical period. These skill scores are calculated as the cumulative minimum
between the observed and modeled distributions of each binned value (Perkins et al., 2007). We
finally select the 6 climate models which exhibit skill scores greater than 0.8, indicating that
these models capture over 80% of the observed probability density functions. The result of this
analysis is shown in Fig. S3, but the interpretation of the results as shown in the main text using
all 23 models does not change.

**3 Results**
**3.1 Rio de Janeiro's spring 2023 heatwave**

The city of Rio de Janeiro experienced exceptionally high temperatures in both the

austral winter and spring of 2023, peaking on November 18 (Fig. 1). This record-breaking event
became the highest daily maximum temperature on record at the Galeão International Airport
weather station, reaching 41.3°C. The extreme heat event was also notable for its accompanying
high specific humidity, which rose alongside temperature in the days leading up to November 18
(Fig. 1b). The combination of elevated temperature and humidity rendered the event a humid
heat extreme, as measured by wet bulb temperature, which peaked at 28.2°C on November 18
(Fig. 1c). The coincidence of extreme dry and wet bulb temperatures is typical for extreme heat
events in Rio de Janeiro, where there is a statistically significant positive correlation between
daily maximum temperature and daily mean specific humidity (Fig. S4). This relationship is
facilitated by the city's abundant access to moisture from the coast and surrounding vegetation.

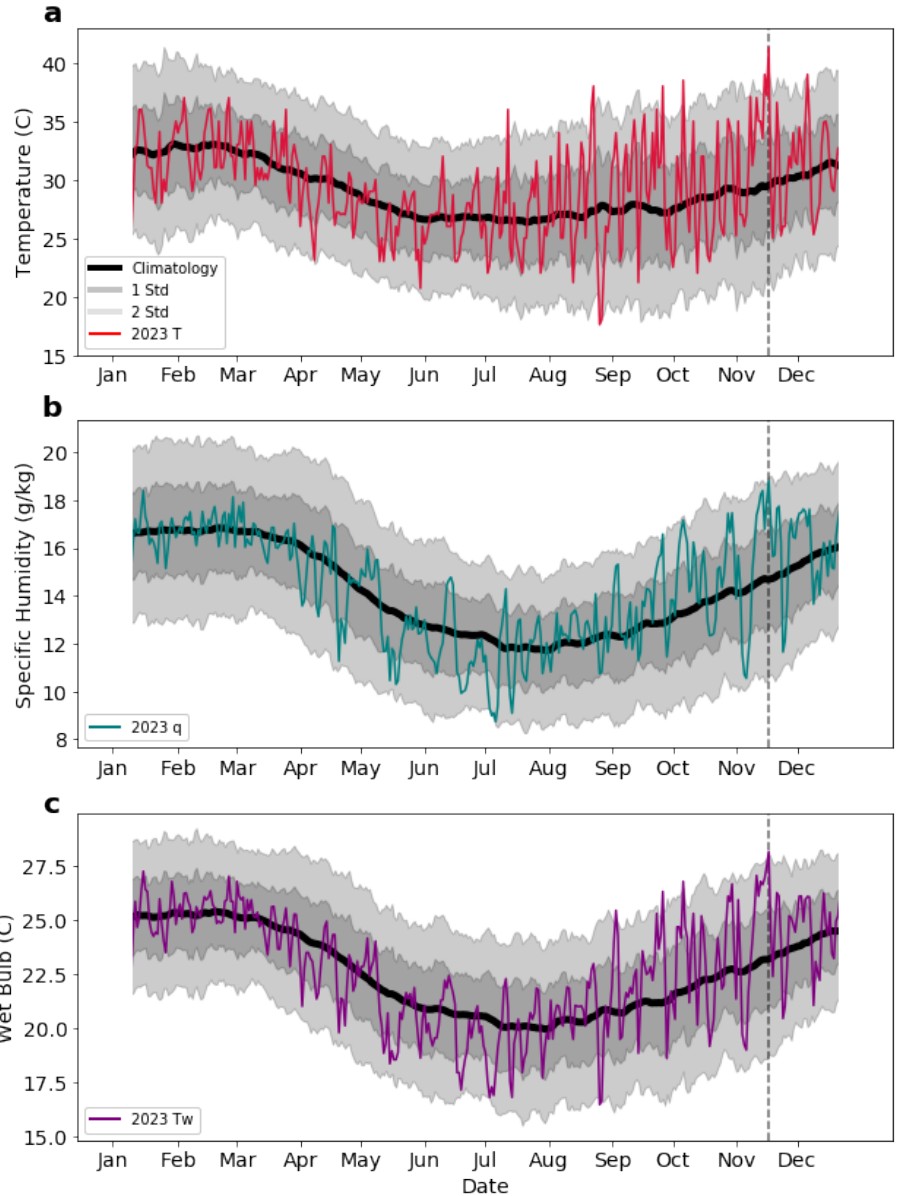

**Figure 1: Historical climatology and 2023 recorded a) daily maximum temperature, b) daily mean specific humidity, and c) daily maximum wet bulb temperature in the city of Rio de Janeiro. Data from the Galeão International Airport weather station as reported by the HadISD dataset. Vertical dashed line identifies record-breaking temperature event on November 18, 2023.**

Elevated temperatures occurred over an area greater than the city of Rio de Janeiro, but were spatially constrained by orography (Fig. 2). We explore the spatial patterns of the heatwave in data from the European Centre for Medium-Range Weather Forecasts (ERA5) reanalysis during the period of 1979-2023 (Hersbach et al., 2020). We see that hotspots in elevated

temperatures were located throughout the coastal region surrounding Rio de Janeiro, with sharp
declines across the mountainous terrain moving inland. These positive coastal temperature
anomalies coincide with northerly surface wind anomalies. ERA5 estimates the daily maximum
temperature on November 18 in the grid cell containing the Galeão International Airport weather
station as 40.6°C, within the range of temperatures recorded throughout weather stations in the
city (Fig. 2c; Fig. S1). This extreme event was also remarkable in length as measured by ERA5,
as daily maximum temperatures were above the locally defined 90th percentile for eight
consecutive days, and above the 99th percentile for the final three days of this period (percentiles
calculated from ERA5 across the period from 1979-2023). This multi-day interval of exceptional
temperatures rendered it difficult for residents to find relief.

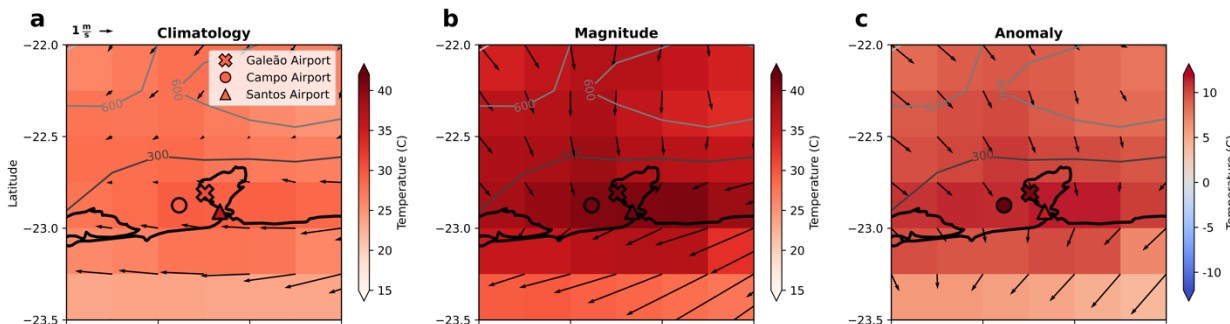


**Figure 2: Spatial maps of daily maximum temperatures during the date of peak extreme**
**heat intensity in the city of Rio de Janeiro using ERA5 data (shading) and three city**
**weather stations with long-term temperature records (markers). Vectors represent surface**
**winds; contours represent elevation in meters. A) Climatology during November 18**
**throughout the historical record. B) Magnitudes on November 18, 2023. C) Anomalies**
**during November 18, 2023.**

The maximum temperature during the event on November 18 coincided in time with
other anomalous meteorological conditions (Fig. 3; for climatological values, see Fig. S5 in the
Supplemental Materials). Positive geopotential height anomalies centered over Rio de Janeiro
were consistent with an intensification of the South American Subtropical High, a semi-
permanent anticyclonic circulation system off the Southeast coast of Brazil. The edge of this
positive high pressure anomaly was collocated with the region of positive temperature anomalies
that includes the city of Rio de Janeiro (Fig. 3b). Surface winds off the coast of Rio de Janeiro
were anomalously northerly. Anomalously northerly flow in this mountainous area can
exacerbate high temperatures directly through downslope winds (Stefanello et al. 2022).
Previous literature has also shown that anomalously northerly winds over the coast can increase
local sea surface temperatures through reductions in wind-driven upwelling, reducing the
capacity for coastal cooling (Castelao and Barth 2006; Palma and Matano 2009). Indeed, positive
SST anomalies of up to 2°C occurred along Rio de Janeiro's coast on the day of the peak in air
temperature (Fig. 3j). Anomalous winds over the interior of South America also enhanced the
northerly South American Low Level Jet (Marengo et al., 2004; Montini et al., 2019). Positive
specific humidity anomalies were present throughout Southeast and South Brazil (Fig. 3f),
intersecting with an area of precipitation along the edge of the low pressure system to the south
(Fig. 3g). The northern portion of the positive specific humidity anomaly was aligned with the
positive geopotential height anomaly off the coast of Southeast Brazil. Widespread negative soil
moisture anomalies occurred throughout most of Brazil, and the interior of South America more
broadly, during this event (Fig. 3d). The large spatial coverage of these negative soil moisture
anomalies was concurrent with Amazonian drought recorded during this time, inherited from the
prior season (Espinoza et al., 2024). These spatial patterns are typical of extreme heat events
during the spring season in the city of Rio de Janeiro, though the magnitudes of the anomalies in
all of these variables are dramatically higher on November 18, 2023 than during other spring
extreme heat events (Fig. 4). The most unique features of the November 18 event were the
intensified northerly winds and the degree of inland penetration of positive specific humidity
anomalies (Fig. 4e-f). Further, the positive local SST anomalies off the coast of Rio de Janeiro
were particularly exceptional in intensity and spatial scale during this event, weakening the sea-
air temperature contrast and sea-breeze (Fig. 4i-j). Outside of these specific distinctions, the
event on November 18, 2023 was an intense example of a typical spring extreme heat event in
the region.

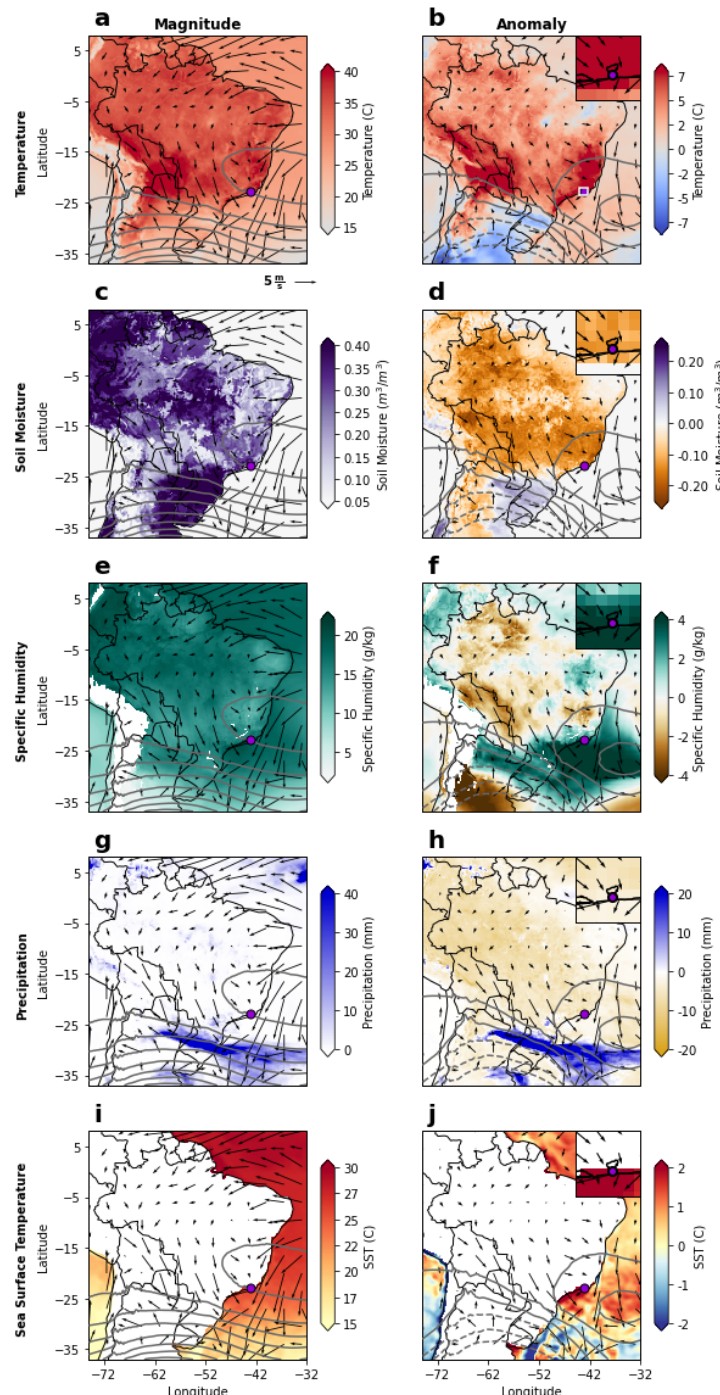


**Figure 3: Magnitude (left) and anomalies (right) of daily maximum temperature, mean soil moisture, mean specific humidity, total precipitation, and mean SST on day of peak temperature in the city of Rio de Janeiro (November 18, 2023). Overlying wind vectors and 500 hPa geopotential height contours (50 m and 25 m contour levels for magnitude and anomaly plots, respectively). Anomalies calculated relative to historical calendar date mean values across the period from 1979-2023. Inset in the upper right corner of each anomaly plot zooms in on the white box surrounding the city of Rio de Janeiro (purple marker) in the top right subplot.**

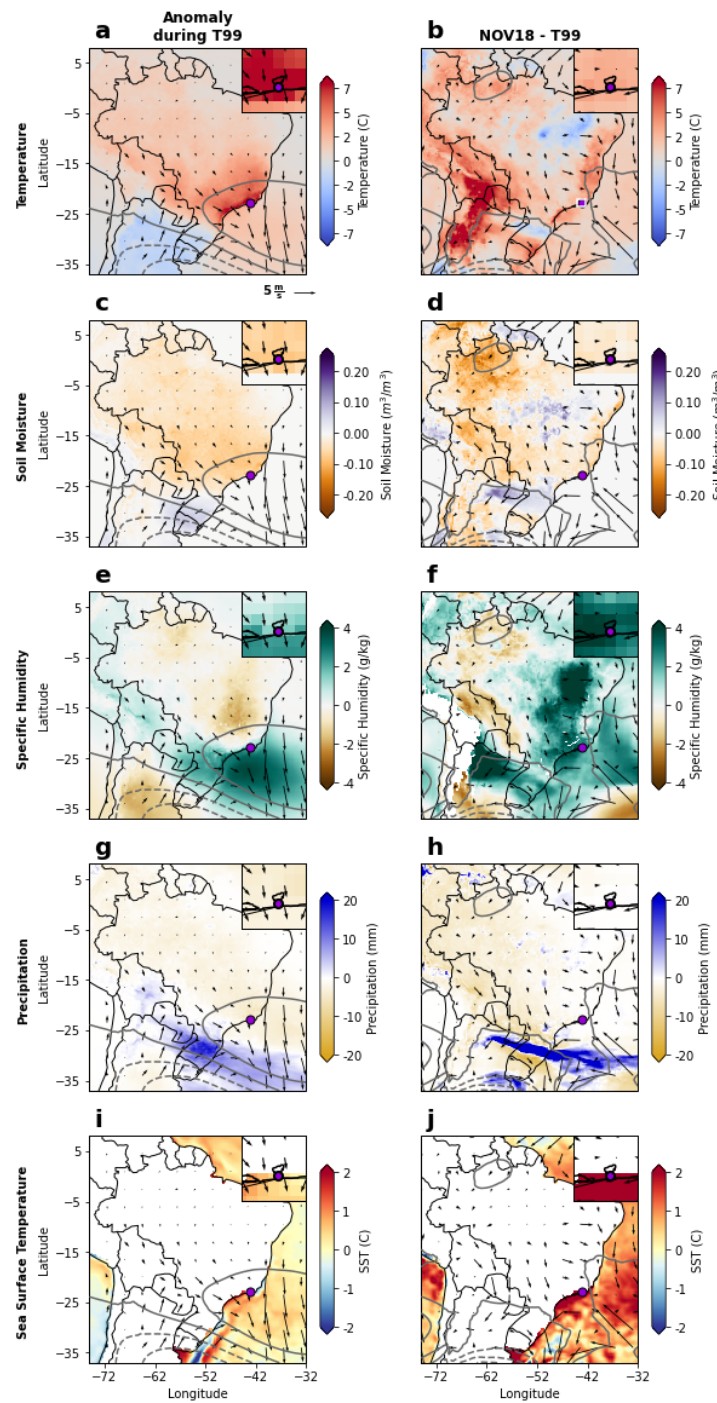

**Figure 4: Anomalies in daily maximum temperature, mean soil moisture, mean specific humidity, total precipitation, and mean SST during 99th percentile extreme temperature days in the September-November (SON) season for the ERA5 grid cell which includes the Galeão International Airport weather station (left). Difference in conditions on November 18 compared to mean conditions during these 99th percentile extreme temperature days (right).**

The time evolution of the meteorological variables described above throughout the month
of November 2023 uncovers the temporal development of the extreme heat event (Fig. 5). Rising
temperatures throughout the weeks leading up to November 18 were preceded by elevated
geopotential heights at 500 hPa and associated atmospheric subsidence. This was accompanied
by a rapid decline in soil moisture which was likely facilitated by the increased solar insolation
associated with the persistent high pressure system and resulting extremely low precipitation
from November 2-November 18. Given that the rainy season in Southeast Brazil typically begins
in late-October to mid-November (Coelho et al., 2021; Latinovic et al., 2018; Marengo et al.,
2012; Liebmann and Mechoso 2011; Raia and Cavalcanti 2008), this period of consecutive dry
days was unusual. Indeed, this period totals 17 days in a row with less than 5 mm of rain per day,
and this only happened during the month of November in one other year in the historical record
from ERA5 between 1979-2023 (2012). These changes in geopotential height, soil moisture, and
suppressed precipitation preceded changes in other variables, evidenced by the grey lines in the
background of each subplot. Wind direction was highly variable on a daily scale, but became
increasingly northerly during this same period. These changes were accompanied by a gradual
increase in SST off the coast of Rio de Janeiro, though delayed compared to that of the local air
temperature. These changes in wind direction and SSTs are likely linked, as upwelling in this
region can be significantly reduced through northerly wind anomalies, increasing coastal sea
surface temperatures (Castelao and Barth 2006; Palma and Matano 2009). Secondary pathways
to SST increases could include increased solar radiation to the ocean, added heat flux to the
ocean, and a thinning of the oceanic mixed layer. These features are common around many
coastlines during atmospheric heatwaves that are associated with warming coastal waters, though
further research would be needed to quantify their relative importance during the November
2023 heatwave in Rio de Janeiro. As air temperatures rose, specific humidity increased over the
city. This was likely related to both local evaporation from the soil (co-occurring with declining
soil moisture) and moisture advected from the anomalously warm coastal waters and surrounding
vegetation. The circulation specifically on November 18 directed wind in the larger region
surrounding Rio de Janeiro to intensify the South American Low Level Jet, which can
additionally increase moisture transport from the Amazon Basin to Southeast Brazil (Marengo et
al., 2004; Vera et al., 2006; Montini et al., 2019). However, convergence of the horizontal
moisture flux at the level nearest the surface was only stronger than its climatological values in
some grid cells within the northern and western areas of the city (Fig. S6). More generally,
specific humidity was also able to build without reaching saturation due to the increasing
temperatures (and the Clausius-Clapeyron relation). Finally, the heatwave was terminated when
a two-day precipitation event occurred from November 19- 20. This precipitation induced a
small decline in specific humidity and SST, as well as a rapid increase in soil moisture.

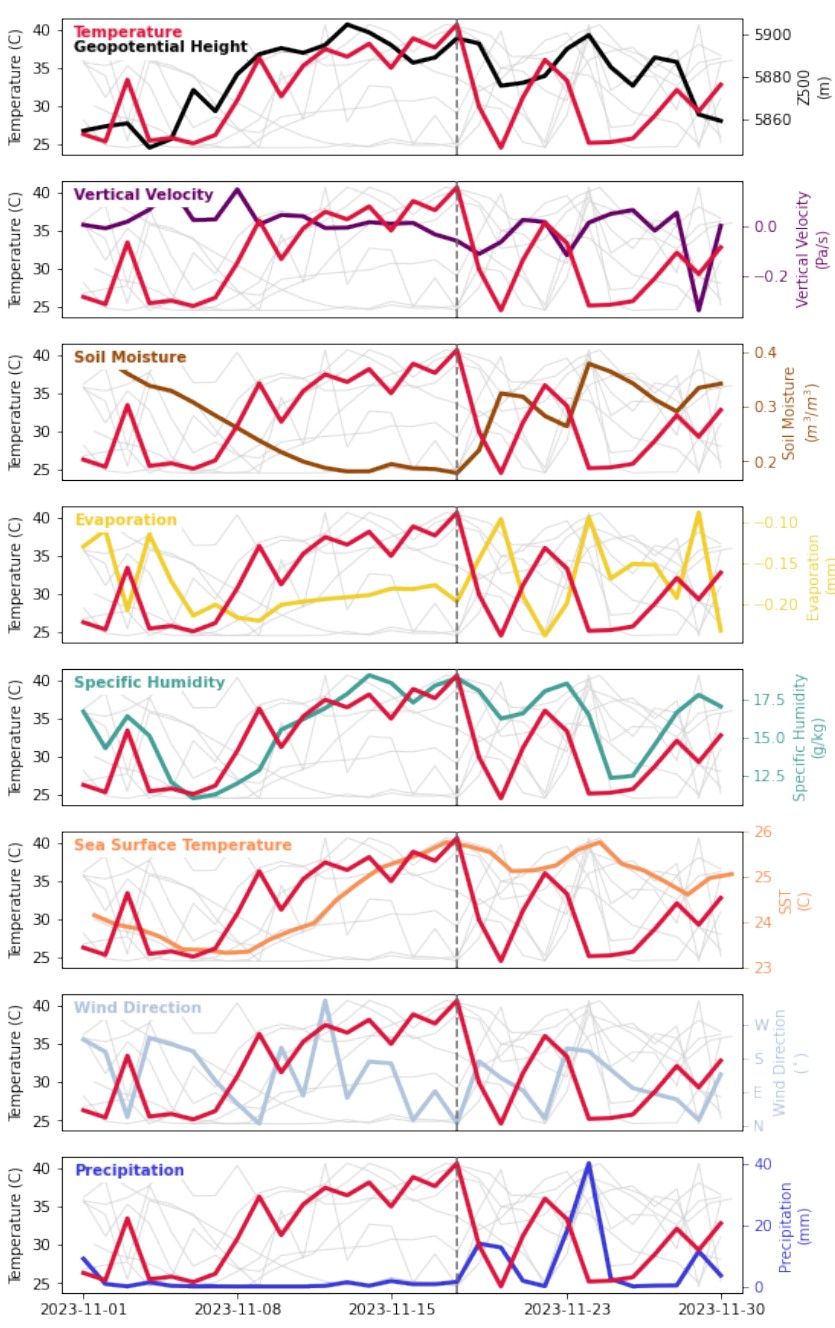


**Figure 5: Evolution of meteorological conditions during the month of November 2023 in Rio de Janeiro. Grey lines in the background of each subplot show the evolution of all variables, with individual variables compared in colors to dry bulb temperature in red. Vertical dashed line identifies record-breaking temperature event on November 18, 2023. All variables are calculated for the grid cell which includes the Galeão International Airport weather station except SST, which is averaged over the box 21°S-24°S and 42°W-45°E.**

The evolution of the 2023 heatwave as shown above is reminiscent of that during the 2010 heatwave analyzed by Geirinhas and coauthors (2019). Those authors explain that the extreme heat event in the summer of 2010 was initiated by a positive SST anomaly over the eastern Pacific that triggered a Rossby wave train that in turn intensified the South Atlantic Subtropical High. Modulation of this climatological high pressure system has been shown to be central to influencing weather in the city of Rio de Janeiro, and particularly temperatures there (Geirinhas et al., 2018). Here we also observe a positive SST anomaly over the equatorial Pacific throughout the month of November and a resulting anomalous wave pattern ending over the South Atlantic High that became increasingly organized and strengthened during the two weeks before November 18 (Fig. 6; see Fig. S7 in the Supplemental Materials for maps of the climatologies and absolute magnitudes of these variables). This mechanism is similar to how El Niño generally influences temperatures in Southeast Brazil on longer timescales (Cai et al., 2020), and we confirm that there is a positive correlation between the ENSO state as quantified by the Niño3.4 index and the frequency of high heat days in the city of Rio de Janeiro in the austral spring season (Fig. S8). Further, the mean spatial SST and Z200 patterns during the spring of typical El Niño years look very similar to those observed during November 2023, though the anomalies in both SST and geopotential height are much larger during November 2023 (Figure S9). 2023 was characterized by a transition from La Niña to El Niño, with the El Niño emerging in April-June 2023 and strengthening to a strong El Niño in the second half of 2023 (Becker et al., 2024). The SST anomalies associated with the El Niño could have been responsible for initiating the wave train which set off the geopotential height anomalies over Rio de Janeiro. More broadly, it has been suggested that multi-month elevations in temperature over Brazil throughout 2023 could be driven in part by El Niño (Pampuch et al. 2025). Further, the second half of 2023 was exceedingly warm globally (Cattiaux et al. 2024; Perkins-Kirkpatrick et al. 2024), due in large part to anthropogenic warming, indicating that climate variability and

climate change both likely preconditioned the November 2023 extreme heat event. We note that
similar wave trains driven by Pacific SST anomalies have been shown to influence weather in
Southeast Brazil even during neutral ENSO states (Seth et al., 2015). Additionally, the
instantaneous extreme temperature event and the preceding persistent dry conditions must also
be linked to the synoptic weather in the area. Decreases in soil moisture and horizontal moisture
fluxes by intensification of the South American Low Level Jet were central features of the
heatwave in 2010, as they were in November 2023. These overlaps in the apparent drivers of the
2010 and 2023 heatwaves underscore that while the 2023 spring event was unprecedented in its
magnitude and unusual in its spring timing, it was not unique in its overall dynamics.

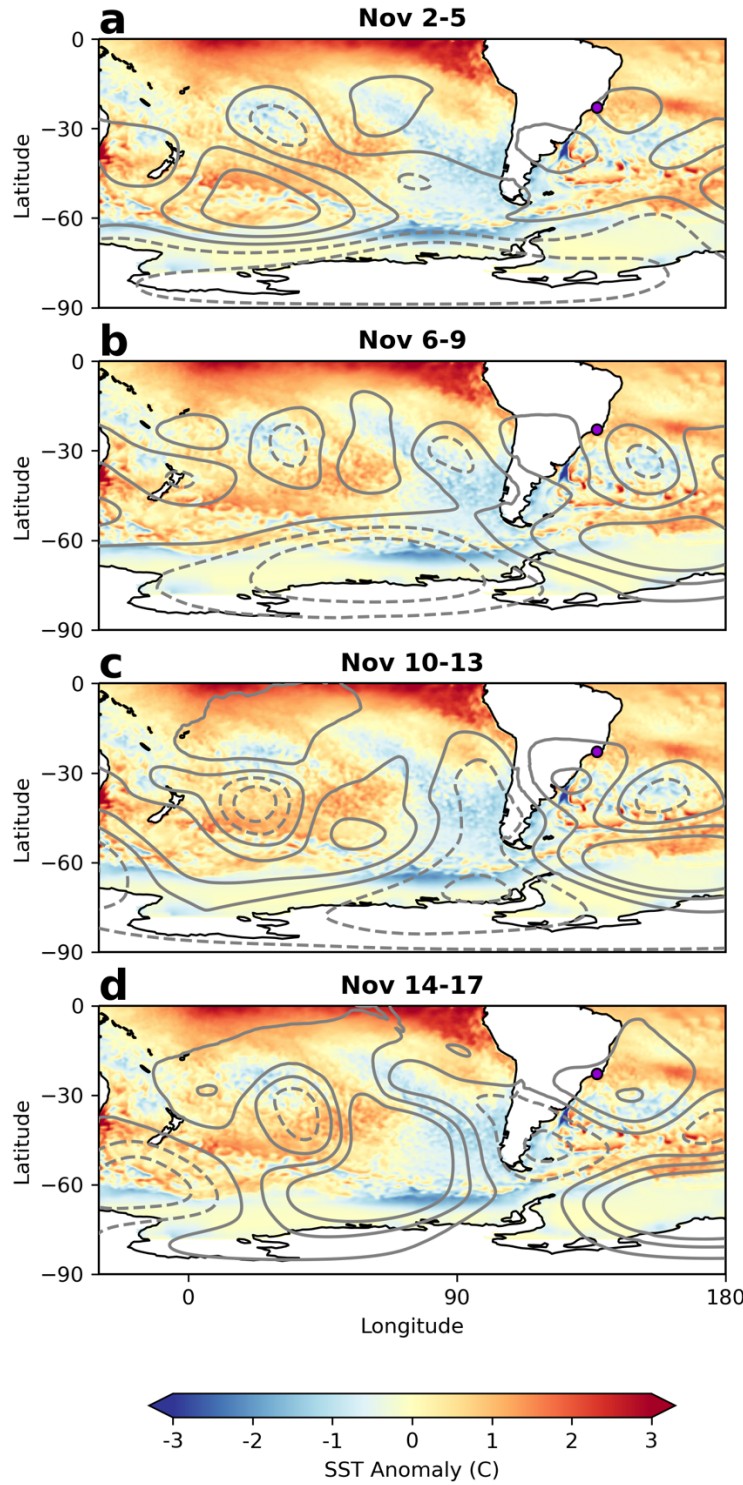


**Figure 6: Evolution of the geopotential height at 200 hPa (contours) in the weeks of suppressed precipitation leading up to the extreme heat event on November 18 in the city of Rio de Janeiro. Geopotential height anomaly contour levels are at 100 m, with positive (negative) anomalies in solid (dashed) contours. Across all subplots, shading indicates November 2023 mean SST anomalies.**

## 3.2 Historical and future changes in extreme heat

Extreme heat events have become more frequent in the city of Rio de Janeiro and the timing of these events has shifted earlier in the calendar year. There has been a significant increase in the number of days above 30°C each year over the past 44 years throughout almost all of South America (Fig. 7a). Further, the number of 90[th] percentile days locally defined at each grid cell has also increased significantly throughout most of the region (Fig. 7b). In the city of Rio de Janeiro specifically, the number of 30°C days per year during the austral spring is increasing at a rate of 0.27 days/year (Fig. 7c). Relative to 1979, the city now experiences almost 12 additional days per year above 30°C during the spring season alone. Overall, the extreme heat season in Rio de Janeiro is broadening. As measured by the number of days between the first and last heatwave day of the season (a period of three or more consecutive days with daily maximum temperatures above 31.4°C), the extreme heat season has lengthened from 156 days in the 1979-1980 season to 176 days in the 2022-2023 season (Fig. 8). The broadening of the heat season is due primarily to more early season heatwave days, while the end date of the heat season has not changed significantly.

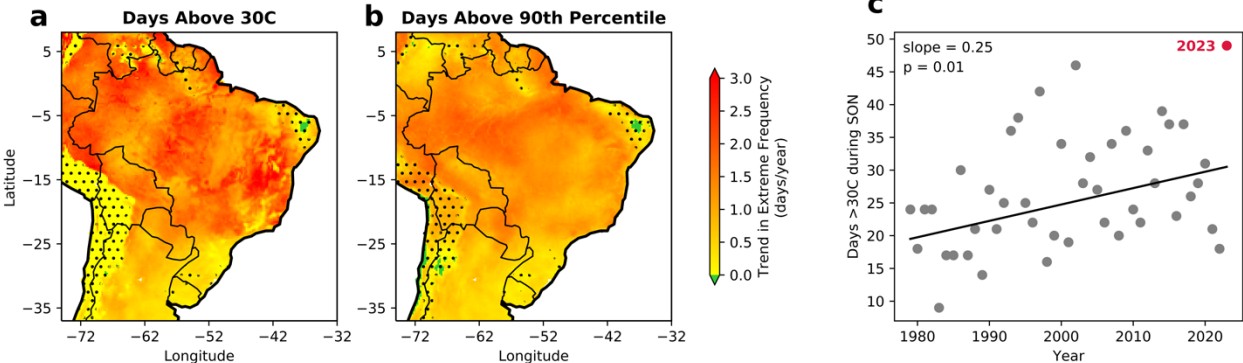

**Figure 7: Historical trend from 1979-2023 in the number of days per year above a) 30°C and b) locally defined 90[th] percentile. Stippling shows areas which are not significant at a p = 0.05 level assessed using a Wald Test. c) Trend in number of days per year above 30°C taking place in the SON season in ERA5 for the grid cell which includes the Galeão International Airport weather station.**

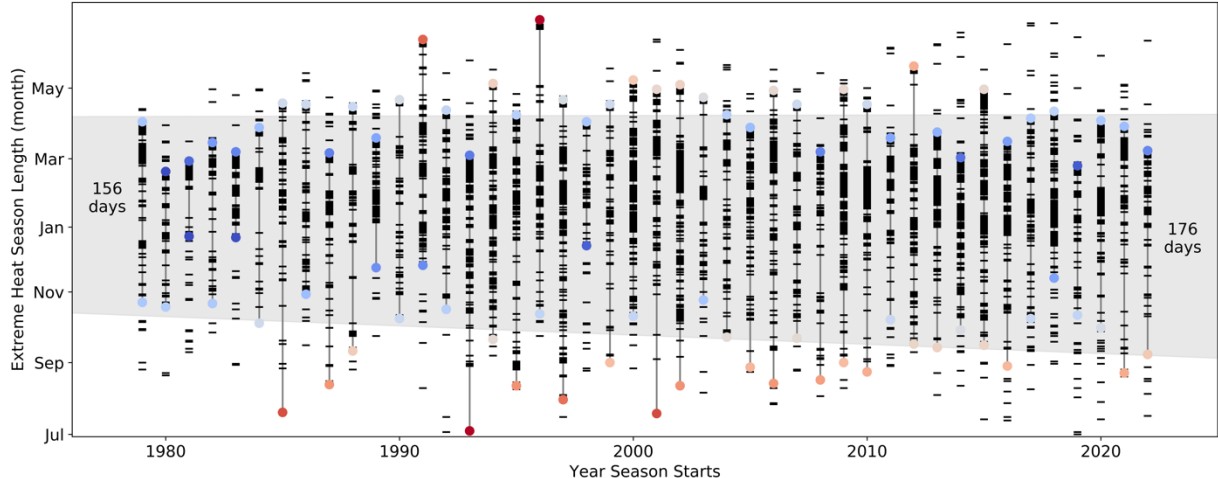

**Figure 8: Shifting timing of the city of Rio de Janeiro extreme heat season. Horizontal axis indicates the year in which winter begins ("January" marking denotes the start of the following calendar year). Colored markers indicate the first and last days of the extreme heat season each year. Marker color indicates whether the start/end date is lengthening (red) or shortening (blue) the heat season compared to historical mean start/end dates. Dashes indicate individual additional days with daily maximum temperatures surpassing 31.4°C (no persistence required). Grey shading indicates area between trend lines in the shifting seasonality.**

The distribution of maximum spring temperatures in the city of Rio de Janeiro has shifted higher over the last four decades and this pattern is projected to continue in the future. In order to evaluate whether the observed historical increase in extreme heat frequency identified in Figures 7 and 8 may continue in the future, we fit annual maximum spring temperatures from historical ERA5 reanalysis data and future projections from bias-corrected NASA Earth Exchange Global Daily Downscaled Projections (NEXGDDP) data (Thrasher et al., 2022, see Methods) to a Generalized Extreme Value (GEV) distribution. When comparing early and late historical periods from 1979-1988 and 2014-2023, respectively, the location parameter of the two GEV distributions has increased by 1.7°C (Fig. 9). The distribution of maximum austral spring temperatures in the city of Rio de Janeiro is also projected to continue shifting to higher values in the future, but the magnitude of this change is strongly dependent upon the future emissions pathway. The temperature distributions associated with mid-century periods (2041-2060) under SSP2-4.5 and SSP5-8.5 future scenarios are similar to that of the last 10 years of observational data, with shifts in the location parameters of 0.1°C and 0.9°C for the two emissions trajectories,

respectively. A larger change is projected by the end of the century (2081-2100) under each
emissions scenario. However, the end-of-century SSP5-8.5 scenario is distinctly separate from
the other distributions, with the distribution location parameter 2.8°C higher than during the last
10 years.

These changes to the distributions strongly influence the probability of an event with the

intensity of the maximum temperature recorded on November 18, 2023. The probability density
function fit to the projected annual maximum spring temperatures under each mid-century period
using a GEV distribution yields a return period for an extreme temperature event with the daily
maximum temperature at least 40.6°C in the city of Rio de Janeiro (analogous to the event on
November 18, 2023 as measured by ERA5) of 51 years under SSP2-4.5 and 33 years under
SSP5-8.5. By the end of the century under either emissions scenario, an event of this magnitude
becomes much more likely, with return periods of 19 years or just 4 years under SSP2-4.5 and
SSP5-8.5, respectively. These return periods align well with estimates for station-level
projections from Collazo et al. 2025, which estimates a return period of 4 to 9 years for
heatwaves analogous to that of November 2023 for a future climate with global mean surface
temperatures 2°C warmer than preindustrial levels. Recent literature has suggested that the
SSP5-8.5 scenario may not be realistic given our current socioeconomic, political, and physical
landscape (Hausfather and Peters 2020; Burgess et al., 2020; Ritchie and Dowlatabadi 2017).
However, these results indicate that an austral spring heatwave of the magnitude experienced in
the city of Rio de Janeiro on November 18 is projected to become much more frequent in the
future, even under the more stringent SSP2-4.5 emission pathway. As temperatures rise and the
city of Rio de Janeiro maintains its ample moisture sources from the nearby ocean and
vegetation, we expect that humid heat extremes will also become more frequent and intense,
though at a slower rate than dry bulb temperature extremes as dictated by tropical atmospheric
dynamics (Coffel et al. 2018; Zhang et al., 2021; Matthews et al., 2025). We must also note that
it is difficult to evaluate whether the models are missing emerging factors that could increase the
frequency and intensity of these extreme heat events – such as Amazonian deforestation or
declines in sea ice – reducing their ability to capture the possible future spring temperature
distributions in Rio de Janeiro.

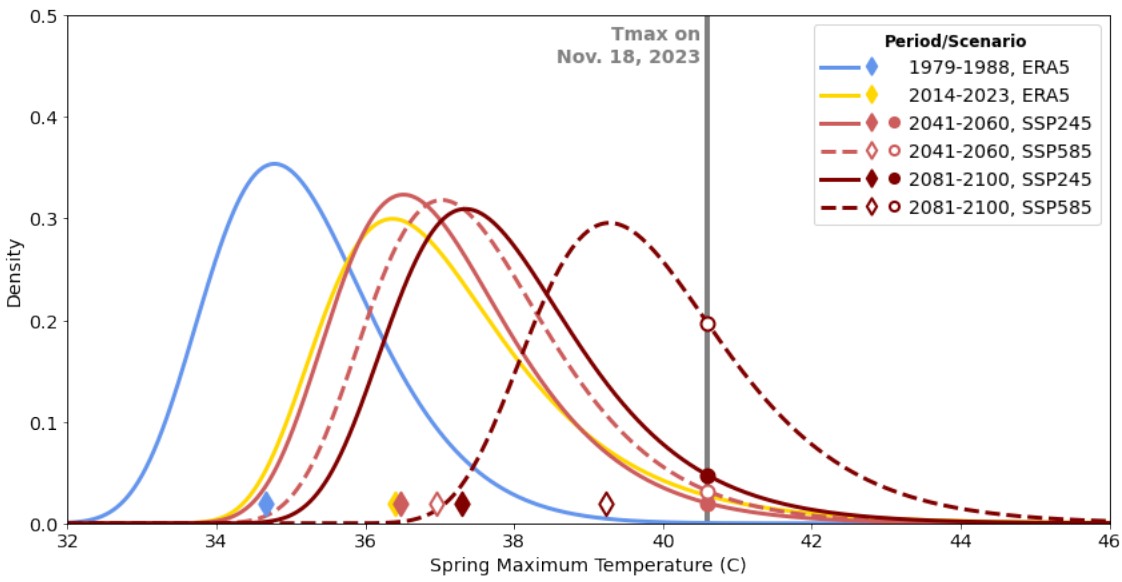


**Figure 9: Generalized Extreme Value distributions for SON maximum temperatures during early and late historical periods (observed in ERA5), mid-century periods, and end-of-century periods under SSP2-4.5 and SSP5-8.5 (projections from bias-corrected NEXGDDP data). Diamonds indicate the value of the location parameter for each distribution. Vertical grey line shows the magnitude of the extreme temperature event on November 18, 2023.**

**4 Conclusions**

The November 2023 heatwave in the city of Rio de Janeiro was a record-breaking event characterized by meteorological conditions largely typical of spring extreme temperature events, but exceptional in their magnitudes. Rising temperatures were associated with positive geopotential height anomalies and corresponding atmospheric subsidence which facilitated clear sky conditions and increased sensible heat flux at the surface. These high pressure anomalies centered over the South Atlantic Subtropical High were likely related to the strong 2023-24 El Niño event. The subsidence near Rio de Janeiro associated with the geopotential height anomalies also suppressed precipitation and facilitated evaporation from the land surface, leading to decreased soil moisture and increased specific humidity. Moisture was available from multiple sources to facilitate these humidity increases, as Rio de Janeiro is a coastal city and downwind of both the Amazon and more local vegetation. SSTs off the coast of Rio de Janeiro were also highly elevated in the days before the heatwave peak, reducing the potential for coastal cooling. Finally, the event was terminated on November 19 due to the evaporative cooling, shading, and mixing associated with the onset of precipitation. The combination of changes in circulation,

land surface feedbacks, and atmosphere-ocean interactions generated the conditions for an
exceptionally intense and persistent extreme heat event in the city of Rio de Janeiro.

The risk of extreme heat in austral spring is increasing significantly in Rio de Janeiro. We

find that extreme spring temperature events are becoming more frequent and that the extreme
heat season is starting earlier and lasting longer than in previous decades. Further, extreme heat
of the magnitude on November 18, 2023 may become much more likely by mid- and end-of-
century periods. However, the absolute increase in the frequency of similar heatwaves is largely
dependent upon our future emissions pathway.

The November 2023 heatwave had devastating impacts, including loss of life. As our

climate continues to change and extreme heat in the city of Rio de Janeiro continues to increase
in intensity and frequency, we can expect more strain on human health and cascading
socioeconomic impacts. This extreme heat event was exceptional not only in its intensity, but
also in its persistence. Consecutive extreme heat days have been shown to have nonlinear
impacts on human health, in Brazil and in other countries, as they prevent individuals, buildings,
and critical electrical equipment from cooling down between heat events (Geirinhas et al., 2020;
Baldwin et al., 2019). More broadly, the direct impacts of heatwaves on hospitalizations
throughout Brazil have been documented, with the largest effects occurring in long duration
events (Zhao et al., 2019). These impacts of heatwaves on mortality are projected to increase,
with particular consequences for elderly populations, especially if targeted adaptation measures
are not put in place (Diniz et al., 2020). Continuing to improve our understanding of how and
when extreme heat occurs is thus essential as our climate continues to change. This is
particularly true for locations such as Rio de Janeiro, which historically has not been a hotspot of
extreme heat – especially in the shoulder seasons – and thus individuals may not be well
acclimated to extreme temperatures then (Periard et al., 2015; Horowitz 2016). The human health
impacts of unusually intense events may be exacerbated by the shifting timing of extreme heat,
as record-breaking exceptional heat events are now occurring outside of the traditional extreme
heat season when individuals may not be prepared to utilize heat mitigation strategies (De Freitas
& Grigorieva, 2015). Because Rio de Janeiro is an area of emerging risk for extreme heat, further
research on models' ability to capture the historical drivers and timing of heatwaves in this
region and evaluations of how these characteristics might shift in the future should be pursued.
The meteorological conditions surrounding the extreme heat event analyzed here
demonstrate the potential for compound hazards throughout Brazil. The identified circulation
pattern that establishes the atmospheric blocking associated with heatwaves in Rio de Janeiro is
also likely linked to heavy precipitation events in South Brazil, an extreme case of which
occurred in May 2024 in center-north of Rio Grande do Sul, including the metropolitan area of
Porto Alegre, displacing hundreds of thousands and killing at least 155 people (Rogero 2024).
The temporal compounding of these extreme temperature and flooding events within Brazil has
the potential to strain the country's disaster management systems more than events occurring in
isolation. Furthermore, the exceptional spatial area within Brazil that experienced anomalous
heat in the November 2023 event, relative to 99[th] percentile heat events in the city of Rio de
Janeiro, underscores the potential for spatially compounding heat that could lead to outsized
impacts. Exploring how unprecedented global surface ocean and surface temperatures, along
with regional features like the broader heat and drought across much of Brazil, may contribute to
extreme heat in the city of Rio de Janeiro will be an important component to improving our
understanding of these compound events' drivers, prediction capacity, and potential to change in
the future.
The evolving meteorological conditions associated with this heatwave were strongly
impacted by the lack of precipitation in the first two weeks of November. This is particularly
unexpected due to the fact that the active phase of the South American Monsoon System
typically begins in late October or early November in this region (Marengo et al., 2012;
Liebmann and Mechoso 2011; Raia and Cavalcanti 2008), which is linked to an increase in
convective activity in tropical South America in the warm season (Jones and Carvalho 2013).
Observational and modeling studies suggest that the South American Monsoon System dry
season is lengthening (Arias et al., 2015; Fu et al., 2013) and that the onset of the active phase is
delaying (Gomes et al., 2022; Pascale et al., 2019). These trends are projected to continue to
some degree in the future with further climate change, particularly in light of ongoing
deforestation which contributes to regional drying trends in the Amazon and other areas of Brazil
(Boisier et al., 2015; Swann et al., 2015). Given Rio de Janeiro is a city with abundant access to
moisture due to its proximity to the coast and vegetation, the increasingly constrained active
monsoon phase could lead to increased frequency and intensity of extreme humid heat in the
spring season (Ivanovich et al., 2024). These changes could be responsible for the evident
asymmetrical historical increase in heat season length during the spring versus fall as
demonstrated here, and extensions of this work should be devoted to an exploration of these
potential relationships.

This work highlights the challenge of analyzing the drivers of weather extremes in such a

climatically diverse city as Rio de Janeiro and emphasizes the need for future research to explore
high resolution comparisons of mechanisms controlling the city's microclimates. Differences
between conditions recorded at individual weather stations within the city's boundaries
demonstrate the degree to which the dynamics of events in each neighborhood depend on the
station's location relative to the coast versus interior (Raymond and Mankin 2019), elevation
(Raymond et al., 2022; Pepin et al., 2015), and degree of urbanization (Kruger et al., 2024;
Chakraborty et al., 2022; Tan et al., 2010). Higher temporal resolution analysis would also better
capture sub-daily processes such as sea breeze and their effect on extreme heat throughout the
city. Further, many of these mechanisms influencing the intracity variability of heat stress
exposure only focus on the effect of differences in dry bulb temperature. Factoring in the spatial
variation in humidity, solar insolation, and windspeed complicate understanding, but are
essential for capturing humans' exposure to heat stress conditions. These intracity differences
also meaningfully impact compound events with non-heat environmental hazards, such as floods,
landslides, droughts, and air pollution, as well as how exposure to these hazards intersects with
areas of social vulnerability. Future work should be devoted to investigating the different
magnitudes of extreme heat and controlling mechanisms throughout Rio de Janeiro in order to
inform targeted extreme heat adaptation plans for individual neighborhoods within the city.

*Data Availability:* The publicly available datasets used in this analysis are accessible via the
following websites: HadISD, https://www.metoffice.gov.uk/hadobs/hadisd/; ERA5,
https://cds.climate.copernicus.eu/datasets/reanalysis-era5-single-levels?tab=overview and
https://cds.climate.copernicus.eu/datasets/reanalysis-era5-pressure-levels?tab=overview; OISST,
https://www.ncei.noaa.gov/products/optimum-interpolation-sst; and NEXGDDP,
https://www.nccs.nasa.gov/services/data-collections/land-based-products/nex-gddp-cmip6.
Station data from the Rio Alert System will be uploaded and accessible via a GitHub repository
upon manuscript publication.

*Code Availability:* All code used for the derivations, calculations, and data visualization will be
made publicly available via a GitHub repository upon manuscript publication.

*Author Contributions:*
S.J.C. conceived of the initial project concept. All co-authors contributed to study design, and
C.I. performed the analysis. C.I. wrote the initial manuscript draft with the feedback and
interpretation of all co-authors. All co-authors read and edited the manuscript.

*Competing Interests:*
The authors declare that they have no conflict of interest.

*Acknowledgements:*
This work was partially supported and funded by Columbia Global at Columbia University,
"Simulation of Extreme Weather Events in Brazilian Megacities", a Climate Hub | Rio Project.
Climate Hub | Rio is a knowledge, research, and innovation hub that brings together experts from
Brazil, Columbia University, and around the world to advance climate-related knowledge and
action in Rio and Brazil. Direct funding for C. Ivanovich and R. Horton was provided by
National Oceanic and Atmospheric Administration's Regional Integrated Sciences and
Assessments program, Grant NA15OAR4310147. A. H. Sobel acknowledges support from NSF
Grant AGS-1933523. S. J. Camargo is partially supported by the NOAA grant
NA23OAR43201600. The authors declare no competing interests.

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
