# Peer review of "Physical Drivers of the November 2023 Heatwave in Rio de"

_EGUsphere, 2025_

## Referee Comment (RC2)

**Broad context and summary of the present study of Ivanovich et al.:**

Specifically in the context of global warming, it is vital to study the drivers of heatwaves and possible future changes in their frequency and intensity. Expanding our understanding is particular important for regions with high population and possibly increased vulnerability.

Fittingly, the authors of this study investigate the drivers of a particularly early and severe heatwave that affected the city of Rio de Janeiro in late November 2023. It is shown that the heatwave was associated with an intensification of the South American Subtropical Anticyclone, possibly linked to a strong El Nino event. The heatwave was possibly further aggravated by a preceding unusual dry spell, high SST temperatures off the coast and anomalously strong northerly winds shutting down sea breeze-related cooling.
The authors extend their study by reviewing possible future changes of heatwaves in the Rio de Janeiro region by means of fitting GEV distributions to bias-corrected projected austral spring temperature extremes, based on a number of different climate models and warming scenarios.

**General comments:**

Overall, this paper constitutes a well-written and valuable case study of this severe heatwave event. The manuscript is well-structured and presents most of the research results in a clear way.
In terms of the scientific content and its novelty, the manuscript could see a few improvements. In my opinion, the authors could have elucidated a bit more deeply whether any particular heatwave driver was particularly significant, as most of the reported anomalies are commonly found in heatwaves outside of the deep tropics (e.g. intensified anticyclones, reduced soil moisture).
While I generally welcome the addition of a section about future heatwave projections, I think that the overall objective of this paper becomes a bit less clear, while at the same time the novelty and depth of each of these topic sections is slightly compromised.
However, I am very positive that this work can be published after some major revision.

**Comments/suggestions about identifying the significance of the multiple heatwave drivers:**

It would be beneficial if the authors could provide some additional analysis into what are the major drivers behind this nearly unprecedented heatwave. Is it really the long dry period and the corresponding increased surface sensible heat fluxes or is the strong northerly wind and its suppression of a sea breeze circulation most important? Or does the northerly wind exacerbate the heat through downslope winds (something that was reported to intensify a summer heatwave in South Brazil in *Stefanello et al., 2022*)?
I understand that it is not easy to produce a physically sound decomposition of the temperature anomaly, such as what has been done within a Lagrangian framework in *Roethlisberger and Papritz (2023)*.
One way to go forward would be to produce a detailed map showing the differences between the maximal heat wave day and the 99[th] percentile composite, something that is missing in the current manuscript (If I am not mistaken, I have only seen heatwave minus climatology, and 99[th] percentile warm days minus climatology, which makes it hard to make out small but possibly important features of that particular heatwave).
In addition, the authors could maybe include a comparison not only against the 99[th] percentile, but to the temporal course of some recent heatwaves of comparable duration and magnitude. To this end, figure 5 could be improved. In its current form, the grey lines in the background are not really helpful, as they are always just showing the non-highlighted other variables.
I would recommend that the authors could show the temporal course of the respective anomalies for

some other 10 recent heatwave events of comparable magnitude.
Maybe the authors could also provide a similar plot to Fig S4, but showing the correlation between near-surface meridional wind and maximum temperatures.

**Comments/suggestions about heatwave drivers in future projections:**

I wonder whether some of the better models, for instance those with a Perkins skill score above 0.8, do also perform well in capturing some of the characteristics of the identified heatwave drivers? If yes and in case these variables are available at a daily scale, it would in my opinion be very valuable to extend the analyses of the future projections by including projections for systematic changes in the suspected heatwave drivers. For instance, one could investigate whether the frequency of days with strong northerly winds is projected to change, or whether there are any substantial changes in the position and strength of the South American Subtropical Anticyclone. A more in-depth analysis into the changes of the suggested main drivers of the exceptional November 2023 heatwave would in my opinion tie the two results sections of the paper closer together and thereby really improve the manuscript.

**Minor comments:**

In general, the text is well-written and does not seem to contain many typos or other minor errors. However, for a further revision, line numbers would be helpful.

The figures are at quite a low resolution.

Figure 8:
Just out of personal curiosity: Are there any ideas about why heatwaves become more frequent in early spring, whereas the end of the heatwave season has not shifted further into early autumn?

Figure 9:
It would be helpful to add, for each scenario, some additional markers for the return level of temperature extremes of the same likelihood of the 2023 heatwave event or something similar.

Figure S8:
The color bar and/or its caption does not seem to be correct for a) and b). Doesn't the plot show a correlation coefficient?

**References:**

Stefanello, Michel, et al. "Spatial–temporal analysis of a summer heat wave associated with downslope flows in southern Brazil: implications in the atmospheric boundary layer." *Atmosphere* 14.1 (2022): 64.

Röthlisberger, Matthias, and Lukas Papritz. "Quantifying the physical processes leading to atmospheric hot extremes at a global scale." *Nature Geoscience* 16.3 (2023): 210-216.

---

## Author Comment (AC1)

**Physical Drivers of the November 2023 Heatwave in Rio de Janeiro**
**Contributions toward Humid Heat**

We sincerely appreciate the careful reviews and helpful suggestions provided by the Reviewers, and thank the Reviewers and the Editor for their time. We have made changes to the manuscript in response to the comments that have considerably enhanced the manuscript. Below, we have provided information on the major modifications to the text and responded point-by-point to comments (reviewer comments in blue, responses in black). When referring to page number X and line number Y, we use the abbreviation PXLY. We reproduce some figures and text from the manuscript for readability, and bolded text indicates revisions to the manuscript.

**Major changes to the paper include:**
- Additional investigation of the relative roles of the extreme heat event's meteorological drivers, resulting in a change to Figure 4.
- Further analysis comparing the circulation and sea surface temperature patterns associated with El Niño to those observed during November 2023, included as new Figure S9.
- Edits to figures for readability (Figure 9, Figure S8).

**Additional changes include:**
- An update to Figure S6 to fix a unit error on the color bar.

**Reviewer 1:**

Using station-based and reanalysis data, the authors present a valuable analysis of the unprecedented November 2023 Rio de Janeiro heatwave, highlighting the driving effect of atmospheric blocking and its linkage to the El Nino event. They also discuss the role of soil moisture deficits associated with the heat extreme event. Finally, they extend the analysis to future projection and find that the frequency of heat extremes is expected to increase in the future. The paper is well-written and present clear results. However, the physical driving mechanism analyses could be further strengthened by quantifying the relative roles of blocking and soil moisture decline in driving the heat extremes. Moreover, the robustness of the linkage between the atmospheric blocking and ENSO needs to be further clarified. I thus recommend a major revision by addressing the following comments.

We thank the Reviewer for these kind comments about the value of our analysis and the compliment on the results' presentation. We appreciate your comments on how to strengthen the claims we make in our result interpretation, and we address these specific suggestions below.

**Specific comments:**
1. One major goal of this study is to figure out the physical drivers of the November 2023 heatwaves. I would like to recommend the authors to perform a thermal budget analysis, which would be helpful to understand the relative contributions of atmospheric circulation and diabatic heating that is mainly influenced by the surface heat fluxes and soil moisture.

We appreciate the Reviewer's suggestion to further explore whether we can quantify the relative contributions of each driver towards the November 2023 extreme heat event. We made a great effort toward executing a thermal budget analysis which followed the methods outlined in Dutra et al. 2017. However, the results of this budget were inconclusive and did not shed light upon the relative contributions of different drivers in causing the extreme heat event in November 2023. We found that the magnitude of each term in the heat budget was highly variable in space and time, and we do not have strong confidence in the reliability of conclusions drawn from such results.

As described throughout the paper, Rio de Janeiro is a very complex city – it is coastal, urban, and mountainous, and these three features change rapidly throughout the city bounds. Because each of these features heavily influences the circulation and diabatic heating, we believe that these extremely fine scale variations would not be well-captured by reanalysis data.

Due to these limitations, we did not include in the manuscript the thermal budget analysis we conducted. However, Reviewer 2 also made suggestions for an alternate analysis to evaluate the relative roles of different extreme heat drivers during the record-breaking event, and we were able to include an additional analysis they suggested. This involved quantifying how much more anomalous each meteorological variable was during November 18, 2023 compared to the mean conditions during 99th percentile extreme heat events in the city. This new analysis clearly identifies the anomalously northerly winds, high sea surface temperatures, and elevated specific humidity as particularly unusual conditions during the extreme heat event.

We have included this analysis in an updated version of Figure 4. Please see our response to Reviewer 2, Comment 2 below for a reproduction of this figure.

2. In the abstract, the authors claim the atmospheric blocking is affected by the El Nino, but they did not provide sufficient evidence for this in the historical analysis. Particularly, they only show the historical correlation between Nino 3.4 index and number of days in heat extreme. I would like to suggest to perform the circulation composite in the spring of El Nino years.

We initially did not include such an analysis because these composites have been shown elsewhere in previous literature (e.g., Cai et al. 2020). However, we agree that the readability of the article could be strengthened by providing a version of this analysis so that it is more readily accessible for readers.

As suggested, we performed a circulation composite of mean geopotential height at 200 hPa and SST anomalies during the spring of El Niño years, reproduced below. This figure agrees with results presented in previous literature (e.g., Cai et al. 2020) and confirms that the mean circulation pattern during the spring of El Niño events is very similar to that observed during the extreme event in November 2023. We have included this final composite as a Supplemental Figure in the same form as the maps in Figure 6, in order to draw attention to the similarities between events. We also add a sentence to the text regarding this figure on P16L347, which reads:

***"Further, the mean spatial SST and Z200 patterns during the spring of typical El Niño years look very similar to those observed during November 2023, though the anomalies in both SST and geopotential height are much larger during November 2023 (Figure S9)."***

[Figure]

Figure S9: Mean SST anomalies (shading) and geopotential height anomalies at 200 hPa (contours) during September-November of El Niño years as defined by the Niño3.4 index. Geopotential height anomaly contour levels are at 8 m, with positive (negative) anomalies in solid (dashed) contours.

**Reviewer 2:**

**General comments:**
1. Overall, this paper constitutes a well-written and valuable case study of this severe heatwave event.

The manuscript is well-structured and presents most of the research results in a clear way. In terms of the scientific content and its novelty, the manuscript could see a few improvements. In my opinion, the authors could have elucidated a bit more deeply whether any particular heatwave driver was particularly significant, as most of the reported anomalies are commonly found in heatwaves outside of the deep tropics (e.g. intensified anticyclones, reduced soil moisture). While I generally welcome the addition of a section about future heatwave projections, I think that the overall objective of this paper becomes a bit less clear, while at the same time the novelty and depth of each of these topic sections is slightly compromised.

However, I am very positive that this work can be published after some major revision.

We greatly appreciate the Reviewer's positive evaluation of the value of this study and the communication of our results. We also are very grateful for the thoughtful comments and thorough review provided by the Reviewer. We address each suggestion in detail below.

**Comments/suggestions about identifying the significance of the multiple heatwave drivers:**
2. It would be beneficial if the authors could provide some additional analysis into what are the major drivers behind this nearly unprecedented heatwave. Is it really the long dry period and the corresponding increased surface sensible heat fluxes or is the strong northerly wind and its suppression of a sea breeze circulation most important? Or does the northerly wind exacerbate the heat through downslope winds (something that was reported to intensify a summer heatwave in South Brazil in *Stefanello et al., 2022*)?

I understand that it is not easy to produce a physically sound decomposition of the temperature anomaly, such as what has been done within a Lagrangian framework in *Roethlisberger and Papritz (2023)*.

One way to go forward would be to produce a detailed map showing the differences between the maximal heat wave day and the 99[th] percentile composite, something that is missing in the current manuscript (If I am not mistaken, I have only seen heatwave minus climatology, and 99[th] percentile warm days minus climatology, which makes it hard to make out small but possibly important features of that particular heatwave).

We appreciate this suggestion from the Reviewer, and we believe that this is a clearer demonstration of how much more extreme the November 2018 heat event was compared to 99[th] percentile temperature events throughout the historical record. Indeed, the most striking differences are the anomalously northerly winds, high sea surface temperatures, and elevated specific humidity. We have edited Figure 4 so that the two columns of this figure are the day-of-year anomaly during T99 events (left) and the difference between conditions during November 18, 2023 and T99 events (right). We have reproduced this updated figure below, for reference.

We also added in a citation of the Stefanello et al. 2022 paper to discuss the potential role of downslope winds in exacerbating the high temperatures, something that was previously not included in our main text. We have also clarified the role of anomalously northerly coastal winds in reducing coastal upwelling in this region. The additional text on P10L254 reads:

*"Surface winds off the coast of Rio de Janeiro were anomalously northerly.* **Previous literature has shown that such anomalously northerly winds over the coast can increase local sea surface temperatures through reductions in wind-driven upwelling, reducing the capacity for coastal cooling (Castelao and Barth 2006; Palma and Matano 2009). Further, anomalously northerly flow in this mountainous area can exacerbate high temperatures directly through downslope winds (Stefanello et al. 2022).***"*

[Figure]

Figure 4: Anomalies in daily maximum temperature, mean soil moisture, mean specific humidity, total precipitation, and mean SST during 99th percentile extreme temperature days in the September-November (SON) season for the ERA5 grid cell which includes the Galeão International Airport weather station (left). Difference in conditions on November 18 compared to mean conditions during these 99th percentile extreme temperature days (right).

3. In addition, the authors could maybe include a comparison not only against the 99th percentile, but to the temporal course of some recent heatwaves of comparable duration and magnitude. To this end, figure 5 could be improved. In its current form, the grey lines in the background are not really helpful, as they are always just showing the non-highlighted other variables. I would recommend that the authors could show the temporal course of the respective anomalies for some other 10 recent heatwave events of comparable magnitude.

Our aim in plotting the other variables in grey behind the main variable of focus in each subplot was to identify the lead/lag between fluctuations in each variable. As a result, we see that geopotential height and soil moisture changed before the other variables of analysis.

However, we also attempted the Reviewer's suggestion to instead plot the evolution of each variable during the two weeks preceding and one week after the 10 hottest days in the historical record (exclusive of being in the same heat event) besides the November 2023 record-breaking event. We have reproduced this version of the figure below, for reference.

However, we find that this version of the figure does not provide new insight regarding differences in the time evolution of this heat event and the associated meteorology – the time evolution of these variables is very similar between the top 10 heatwaves and the November 2023 event. The difference in magnitude during the record-breaking event is what really stands out, but this is already communicated in other figures such as Figure 4. Because of this, we choose to retain the version of Figure 5 from the original manuscript.

To address the Reviewer's comment, we clarify our goal in plotting the evolution of all variables in the background of each subplot in grey on P14L306 with the following additional text:

***"These changes in geopotential height, soil moisture, and suppressed precipitation preceded changes in other variables, evidenced by the grey lines in the background of each subplot."***

[Figure]

Figure: Edited version of Figure 5 as suggested by the Reviewer, with evolution of variables surrounding 10 hottest heat events plotted in each subplot background in grey.

4. Maybe the authors could also provide a similar plot to Fig S4, but showing the correlation between near-surface meridional wind and maximum temperatures.

Thank you for this suggestion. We created the suggested plot, and there is a significant negative correlation between meridional wind and surface temperature, but it is very weak. Accordingly, we do not include this additional figure in the manuscript.

[Figure]

Figure: Correlation between daily maximum temperature and daily mean meridional wind in Rio de Janeiro for a) full year and b) SON season. Data measured by the Galeão International Airport weather station as reported by the HadISD dataset.

**Comments/suggestions about heatwave drivers in future projections:**
5. I wonder whether some of the better models, for instance those with a Perkins skill score above 0.8, do also perform well in capturing some of the characteristics of the identified heatwave drivers? If yes and in case these variables are available at a daily scale, it would in my opinion be very valuable to extend the analyses of the future projections by including projections for systematic changes in the suspected heatwave drivers. For instance, one could investigate whether the frequency of days with strong northerly winds is projected to change, or whether there are any substantial changes in the position and strength of the South American Subtropical Anticyclone. A more in-depth analysis into the changes of the suggested main drivers of the exceptional November 2023 heatwave would in my opinion tie the two results sections of the paper closer together and thereby really improve the manuscript.

We appreciate this suggestion by the Reviewer to explore how well models with a relatively high Perkins skill score are able to capture the heatwave characteristics. However, we are unable to do so due to limitations of the downscaled product we use to generate the projections used in this paper. The NEXGDDP dataset only possesses a small number of downscaled variables, and surface wind direction, sea surface temperature, soil moisture, and geopotential height are all unavailable in this dataset. Because these variables are not accessible, we cannot explore the same heatwave drivers as in the observational analysis to determine how well they are represented.

We believe that a separate study could be devoted to 1) evaluating models' ability to capture these heatwave characteristics and 2) investigating how these drivers are expected to shift in the future. However, this is beyond the scope of the present analysis. We have included a sentence motivating this future research topic in the conclusion on P22L483, which reads:

*"The human health impacts of unusually intense events may be exacerbated by the shifting timing of extreme heat, as record-breaking exceptional heat events are now occurring outside of the traditional extreme heat season when individuals may not be prepared to utilize heat mitigation strategies (De Freitas & Grigorieva, 2015). Because Rio de Janeiro is an area of emerging risk for extreme heat, further research on models' ability to capture the historical drivers and timing of heatwaves in this region and evaluations of how these characteristics might shift in the future should be pursued."*

We also acknowledge the Reviewer's comment that the sections of the paper focused on historical versus future heatwaves feel disjointed. We view the two paragraphs evaluating the historical and future frequency of heatwaves like the one in November 2023 as a natural final analysis of the study, serving as an extension of the trend analyses presented in the previous paragraph of the section. We show how the frequency of heat extremes has increased over the historical period (Figure 7 and 8) and then follow this with an evaluation of how extremes are expected to change in the future (Figure 9). We recognize that this structure was insufficiently clear, and we have added a sentence to help guide the reader through the progression of the analysis on P19L402, reading:

*"The distribution of maximum spring temperatures in the city of Rio de Janeiro has **shifted higher** over the last four decades and **this pattern is projected** to continue in the future. **In order to evaluate whether the observed historical increase in extreme heat frequency identified in Figures 7 and 8 may continue in the future,** we fit annual maximum spring temperatures from historical ERA5 reanalysis data and future projections from bias-corrected NASA Earth Exchange Global Daily Downscaled Projections (NEXGDDP) data (Thrasher et al., 2022, see Methods) **to** a Generalized Extreme Value (GEV) distribution."*

**Minor comments:**

6. In general, the text is well-written and does not seem to contain many typos or other minor errors. However, for a further revision, line numbers would be helpful.

Thank you for this note, and we completely agree that this is helpful for the review process. While our PDF had line numbers upon upload, the preprint website *In Review* removes them upon posting. If we are able to include them in our revision, we will.

7. The figures are at quite a low resolution.

This is also unfortunately a feature of the preprint website, which will be improved upon for the final upload of the manuscript.

8. Figure 8: Just out of personal curiosity: Are there any ideas about why heatwaves become more frequent in early spring, whereas the end of the heatwave season has not shifted further into early autumn?

This is a wonderful question, and one we are exploring in a present research project. We believe that the asymmetry in the increase of heatwave frequency in the spring versus autumn is due to

the trend in South American Monsoon System precipitation, which prevailing literature suggests is starting later but not extending later (Arias et al. 2015; Fu et al. 2013; Gomes et al. 2022; Pascale et al. 2019). If suppressed precipitation serves to increase the likelihood for extreme heat events as we see in November 2023, these changes in the timing of monsoon precipitation would allow temperatures to rise predominantly in the spring.

The suggestion for future research is included in the Conclusions section starting on P23L506, but we have added further context about how this hypothesis relates to the results we outline in the paper. We reproduce this section below:

*"The evolving meteorological conditions associated with this heatwave were strongly impacted by the lack of precipitation in the first two weeks of November. This is particularly unexpected due to the fact that the active phase of the South American Monsoon System typically begins in late October or early November in this region (Marengo et al., 2012; Liebmann and Mechoso 2011; Raia and Cavalcanti 2008), which is linked to an increase in convective activity in tropical South America in the warm season (Jones and Carvalho 2013). Observational and modeling studies suggest that the South American Monsoon System dry season is lengthening (Arias et al., 2015; Fu et al., 2013) and that the onset of the active phase is delaying (Gomes et al., 2022; Pascale et al., 2019). These trends are projected to continue to some degree in the future with further climate change, particularly in light of ongoing deforestation which contributes to regional drying trends in the Amazon and other areas of Brazil (Boisier et al., 2015; Swann et al., 2015). Given Rio de Janeiro is a city with abundant access to moisture due to its proximity to the coast and vegetation, the increasingly constrained active monsoon phase could lead to increased frequency and intensity of extreme humid heat in the spring season (Ivanovich et al., 2024). **These changes could be responsible for the evident asymmetrical historical increase in heat season length during the spring versus fall demonstrated here, and** extensions of this work should be devoted to an exploration of these potential relationships."*

9. Figure 9: It would be helpful to add, for each scenario, some additional markers for the return level of temperature extremes of the same likelihood of the 2023 heatwave event or something similar.

We thank the reviewer for this suggestion, which we believe makes it easier to visualize the order of return periods associated with the various future time periods and emissions scenarios.

Note we have also rewritten the date label in grey in a more globally consistent way (rather than MM/DD/YYYY).

[Figure]

Figure 9: Generalized Extreme Value distributions for SON maximum temperatures during early and late historical periods (observed in ERA5), mid-century periods, and end-of-century periods under SSP2-4.5 and SSP5-8.5 (projections from bias-corrected NEXGDDP data). Diamonds indicate the value of the location parameter for each distribution. Vertical grey line shows the magnitude of the extreme temperature event on November 18, 2023.

10: Figure S8: The color bar and/or its caption does not seem to be correct for a) and b). Doesn't the plot show a correlation coefficient?

Thank you for catching this error, which is a leftover colorbar label from the similarly structured plot in Figure 7. We have updated the colorbar accordingly to read, "Correlation Coefficient."

[Figure]

Figure S8: Historical correlation from 1979-2023 between Niño3.4 index and number of days per year above a) 30°C and b) locally defined 90th percentile. Stippling shows areas which are not significant at a p = 0.05 level. c) Correlation between the number of days above 30°C per year taking place in the SON season in ERA5 for the grid cell which includes the Galeão International Airport weather station and the Niño3.4 index.

**Reviewer Response Document References**

Stefanello, M., Ewerling da Rosa, C., Bresciani, C., Cordero Simões dos Reis, N., Stefanello Facco, D., Teleginski Ferraz, S. E., et al., (2022). Spatial–temporal analysis of a summer heat wave associated with downslope flows in southern Brazil: implications in the atmospheric boundary layer. Atmosphere, 14(1), 64. https://doi.org/10.3390/atmos14010064

Dutra, L. M. M., da Rocha, R. P., & Lee, R. W. (2017). Structure and evolution of subtropical cyclone Anita as evaluated by heat and vorticity budgets. Quarterly Journal of the Royal Meteorological Society, 143(704), 1539-1553. https://doi.org/10.1002/qj.3024

---

## Author Response (AR3)

**Physical Drivers of the November 2023 Heatwave in Rio de Janeiro**

Catherine Ivanovich, Adam Sobel, Radley Horton, Ana Nunes, Rosmeri Rocha, and Suzana Camargo

**Editor's Comments**

Dear Catherine, many thanks. I am satisfied with the responses and new manuscript. For completeness, I think you may decide to cite the following paper https://www.frontiersin.org/journals/climate/articles/10.3389/fclim.2025.1529082/full in which the large scale conditions behind the heat waves of winter and spring 2023 are discussed. I kindly ask you to see if this paper is appropriate to support your results and reference it appropriately. Then, I'll be ready to accept the manuscript for publication.

We are glad to hear that the Editor is satisfied with the changes we made to the last draft of the manuscript. We appreciate them bringing the paper above to our attention, which we have included as a citation in our manuscript when discussing the effects of large scale circulation patterns potentially associated with El Niño on the November 2023 heatwave in Rio de Janeiro. The new text citing this paper starting on P16L363 reads (new text in bold):

*"The SST anomalies associated with the El Niño could have been responsible for initiating the wave train which set off the geopotential height anomalies over Rio de Janeiro, **consistent with the results of a recent analysis exploring spring and winter heatwaves throughout South America during 2023 (Marengo et al. 2025).**"*